



Comment on "Isotopic evidence for dominant secondary production of HONO in
near-ground wildfire plumes."
*James M. Roberts, Chemical Sciences Laboratory, NOAA/ESRL, Boulder, CO.,*
**Abstract**
Chai et al. recently published measurements of wild fire (WF) derived oxides of nitrogen (NOx)
and nitrous acid (HONO) and their isotopic composition. The method used to sample NOx,
collection in alkaline solution, has a known 1:1 interference from another reactive nitrogen
compound, acetyl peroxynitrate (PAN). Although PAN is thermally unstable, subsequent
reactions with nitrogen dioxide ($NO_2$) in effect extend the lifetime of PAN many times longer
than the initial decomposition reaction would indicate. This, coupled with the rapid and efficient
formation of PAN in WF plumes, means the NOx measurements reported by Chai et al. were
severely impacted by PAN. In addition, the model reactions in the original paper did not include
the reactions of $NO_2$ with hydroxyl radical (OH) to form nitric acid, nor the efficient reaction of
larger organic radicals with nitric oxide to form organic nitrates ($RONO_2$).

**Main Text**
Chai et al., (2021) present ground-based measurements of nitrous acid (HONO) and the
oxides of nitrogen (NOx) and their $^{15}N$ and $^{18}O$ isotopic abundances in airmasses in the
immediate vicinity of wildfire (WF) (Chai, et al., 2021a). Although the paper was openly
reviewed by me (Roberts, 2021) and somewhat modified as a result, there are several aspects of
the methodology and interpretation of the results that bear further commenting on.
The main problem with the methodology, sample collection of NOx in alkaline solution,
is a 1:1 interference (as N) from acetyl peroxynitrate ($CH_3C(O)OONO_2$, PAN) a ubiquitous
product of volatile organic compound (VOC)-NOx photochemistry. We have known this for
decades as alkaline collection and hydrolysis has been used in a number of studies as a means to
calibrate PAN sources (see for example Stephens, 1969, Grosjean et al., 1984, Grosjean and
Harrison, 1985, and references in Roberts 1990). The basis of PAN interference with NOx
collection is really not in dispute since PAN is at least 2-orders of magnitude more soluble in
water than NO and $NO_2$ (Sander, 2015) and its alkaline hydrolysis is extremely rapid and forms



nitrite (see for example Steer et al., 1969). Nitrite of course is one main product of the aqueous
reactions of NOx and is converted to nitrate by permanganate in the method of Chai et al.,
(2021a). Thus, any conditions that will completely collect and convert NOx to nitrate will
convert PAN to nitrate quantitatively.

36        Regrettably, there are no PAN measurements in this study with which to constrain this

interference. However, Chai et al., argue in their reply and in their paper (Chai et al., 2021a&b),
that very little PAN would be present in the air masses they sampled due to the relative freshness
of the WF emissions not having made much PAN yet and/or due to the thermal instability of
PAN. The gist of this Comment is that the arguments underlying two assumptions are simply not
correct.

42        The kinetics of the thermal decomposition of PAN and related compounds are well

studied, (see for example the review by Kirchner et al., (1999), and the more recent work of
Kabir (2014)). It is true that the initial reaction:

46        $CH_3C(O)OONO_2 + M \rightarrow CH_3C(O)OO + NO_2$                                     (1)


is relatively rapid at 298K, $4 \times 10^{-4}$ s$^{-1}$. However, net loss of PAN only occurs if the peroxy
radical is lost due to reaction with nitric oxide;

51        $CH_3C(O)OO + NO \rightarrow CH_3 + CO_2 + NO_2$                                     (2)


The other main fate of the peroxyacetyl (PA) radical is recombination with $NO_2$ to reform PAN

55        $CH_3C(O)OO + NO_2 \rightarrow CH_3C(O)OONO_2$                                     (3)


As a result, the loss of PAN due to thermal decomposition should be expressed as an effective
net loss rate;

60        Net Loss Rate $= k_1\{1-(k_3[NO_2]/(k_3[NO_2] + k_3[NO]))\}$                                     (4)




This introduces a dependence of the loss rate on the ratio [NO₂]/[NO] shown in Fig. 1. NOx
measurements from the Chai study itself correspond to lifetimes as long as 8 hours, clearly long
enough for PAN to persist at ground level, and to be mixed throughout the daytime planetary
boundary layer (PBL). This aspect of PANs chemistry has considerable experimental support,
(see for example Roberts et al., 2007), and is one of the reasons why PANs are universally
observed at ground level. In fact, the only instances in which ground level PAN concentrations
over the continent reach very low levels, essentially zero, are nighttime periods under very stable
nocturnal boundary layers with local sources of nitric oxide (see for example Roberts et al.,
2002). Daytime PAN levels are more characteristic of the entire daytime PBL because of the
relatively long lifetime and PBL mixing times on the order of a few hours. The other major gas
phase losses of PA radical involve reaction with $HO_2$ or $RO_2$ radicals and have similar rate
constants to Reactions (2&3), 1- $2 \times 10^{-11}$ $cm^3 molec^{-1} s^{-1}$, so will only be important when NOx
levels are below 100 pptv or so. It should also be noted that other photochemical loss processes
for PAN (photolysis, reaction with OH) are quite slow compared to thermal decomposition
(Talukdar et al., 1995) and will not impact PAN in this environment. So, it is possible that that
the Chai et al., (2021a) results from young nighttime plumes could have been had very low to no
PAN contributions, but all the other sampling instances will have had substantial contributions
due to PAN.
The magnitude of this contribution could be quite large due to rapid a formation of PANs
in WF plumes and the fact that long PAN lifetimes mean that formation of PANs aloft will
impact the ground due to mixing. As indicated by the references quoted by Chai et al, (2021a)
and other relatively recent papers, PANs formation is often rapid in WF plumes, so PAN/NOx
can reach or exceed 1 within ½ -to-several hours. For example: Alvarado et al., (2010) observed
40% conversion of WF NOx to PAN within a few hours, Juncosa Callahorrano (2020) found
PAN/NOx averaged more than 1 after 1 hr of processing during one of the same projects as Chai
et al. So, we can conclude with considerable confidence that production of PAN from WF NOx
likely impacted airmasses that authors have termed as "young daytime", and certainly impacted
the airmasses that have been categorized as "mixed" and "aged".
Chai et al. (2021a&b) also use the results of their [15]N-NOx measurements, particularly
those of the aged categories, to argue against a significant contribution from PAN. However,
their conceptual model does not take into account the other NOx chemistry that we know to be





taking place in this environment: reaction of $NO_2$ to form $HNO_3$ from $OH + NO_2$, the formation
of organic nitrate species $RONO_2$ from reaction of $RO_2$ radicals with NO, and nighttime
reactions of $NO_3$ and $N_2O_5$. So, PAN formation is not the only reactive nitrogen chemistry that
will shift $\delta^{15}N$-NOx (which is actually $\delta^{15}$-(NOx + PAN) to lower values relative to the WF
signature. This other NOx chemistry needs to be considered.

**Competing Interests**
The author declares no competing interests.

**Acknowledgments**
This work was supported by NOAA's Climate research and Health of the Atmosphere
Initiative.

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



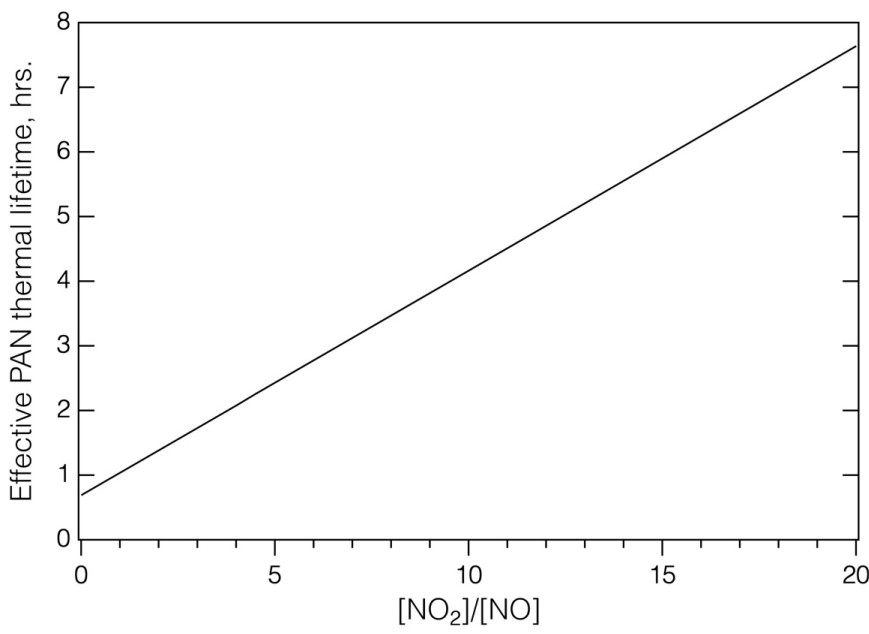


Figure 1. The effective PAN thermal lifetime at 298K as a function of [NO$_2$]/[NO] based on
Reactions (1-3).