# Peer review of "Comment on "Isotopic evidence for dominant secondary production of HONO in"

_Atmospheric Chemistry and Physics, 2021_

## Community Comment (CC1)

**Comment on "Comment on "Isotopic evidence for dominant secondary production of HONO in near-ground wildfire plumes.""**

Jiajue Chai[1,2], Jack E. Dibb[3], Bruce E. Anderson[4], Claire Bekker[1,2], Danielle E. Blum[1,5], Eric Heim[3], Carolyn E. Jordan[4,6], Emily E. Joyce[1,2], Jackson H. Kaspari[7], Hannah Munro[3], Wendell W. Walters[1,2], and Meredith G. Hastings[1,2]

[1] Institute at Brown for Environment and Society, Brown University, Providence, RI

[2] Department of Earth, Environmental and Planetary Sciences, Brown University, Providence, RI

[3] Institute for the Study of Earth, Ocean and Space, University of New Hampshire, Durham, NH

[4] NASA Langley Research Center, Hampton, VA

[5] Department of Chemistry, Brown University, Providence, RI

[6] National Institute of Aerospace, Hampton, VA

[7] Department of Chemistry, University of New Hampshire, Durham, NH

*Correspondence to*: Jiajue Chai (jiajue_chai@brown.edu)

We thank Dr. James Roberts for sharing his expertise on PAN. We acknowledge that at significant concentrations of PAN, i.e., comparable to that of $NO_x$ in the atmosphere, and PAN could be efficiently collected in the permanganate impinger solution, it would interfere with the $NO_x$, also collected as nitrate, for isotopic analysis. It is uncertain whether significant PAN exists in the ground environments where we conducted our sampling, because no direct near-ground PAN concentration measurements in BB plume impacted areas are available. However, the isotopic results can shed unique light on whether PAN interference is important in our case. For aged smoke, we would expect $\delta^{15}N$-$NO_x$ to decrease from that in fresh emissions due to partial transformation of $NO_x$ to additional oxidized N products (e.g. PAN), as well as isotopic exchange between $NO_x$ and these oxidized species; both processes will leave $^{15}N$ depleted in $NO_x$ (relative to $^{14}N$) and $^{15}N$ enrichment in PAN (Walters and Michalski, 2015). If PAN existed at significant concentrations that were 1) comparable with $NO_x$ in the atmosphere, and 2) completely collected in the permanganate solution, then the $\delta^{15}N$-$NO_3^-$ would reflect the overall $\delta^{15}N$ of $NO_x$ + PAN in the final reduced permanganate solution. In this case, we would expect that aged smoke would not shift from the $\delta^{15}N$-$NO_x$ range of young smoke, because $\delta^{15}N$ shifts in both PAN and $NO_x$ could offset each other. However, our observed $\delta^{15}N$-$NO_x$ mean values for both aged daytime and nighttime smoke are significantly ($p<0.05$) lower than that of the young smoke (shown in the figure below). This $^{15}N$ depletion in collected "$NO_x$" indicates the $NO_x$ in aged smoke was the predominant N species collected in the permanganate impinger during our field campaign. Similar analysis was also discussed by Miller et al. (2017). Therefore, we do not find significant isotopic evidence that PAN interferes with $NO_x$ for $\delta^{15}N$ characterization in our study conditions.

[Figure]

[Figure]

Additionally, $OH+NO_2/NO+RO_2$ and $NO_2+NO_3$ are expected to slightly deplete $^{15}N$ in $NO_x$, due to their recombination reaction characteristics that have a secondary isotope effect. Our isotopic mass balance model was constructed to predict the $\delta^{15}N$ difference between $NO_x$ and secondarily produced HONO. Within the timescale of HONO destruction (especially during the daytime), the $\delta^{15}N$ difference resulting from each of the HONO production reactions (4‰-40‰) is way larger than the change of $\delta^{15}N$-$NO_x$ itself (<0.3‰) resulting from either $NO_2$-to-HONO conversion or $OH+NO_2/NO+RO_2$. As such, it is reasonable to consider that the change of $\delta^{15}N$-$NO_x$ itself has negligible impact on our modeling results.

In conclusion, our direct isotopic evidence does not show a significant PAN interference in the $NO_x$ collected for isotopic analysis. However, it is interesting to see the contrast between our results and Dr. Roberts' expectation. As such, a direct PAN concentration measurement is really needed for future near-ground measurement of smoke plumes. In addition, a validated PAN collection method for isotopic analysis will also be helpful.

References:

Miller, D. J., Wojtal, P. K., Clark, S. C., and Hastings, M. G.: Vehicle NOx emission plume isotopic signatures: Spatial variability across the eastern United States, J. Geophys. Res. Atmospheres, 122, 4698–4717, https://doi.org/10.1002/2016JD025877, 2017.

Walters, W. W. and Michalski, G.: Theoretical calculation of nitrogen isotope equilibrium exchange fractionation factors for various NOy molecules, Geochim. Cosmochim. Acta, 164, 284–297, https://doi.org/10.1016/j.gca.2015.05.029, 2015.

---

## Community Comment (CC2)

**Dear Dr. Jaffe,**

Thank you for your helpful comments on our paper!

Indeed, PAN can cause interference to the  $NO_x$  in the alkaline collection system, if PAN exists in significant proportion relative to NOx. However, as we pointed out in our response to Dr. Roberts' comments, our direct isotopic evidence does not show a significant PAN interference in the NOx collected for isotopic analysis. For aged smoke, we would expect  $\delta^{15}$ N-NOx to decrease from that in fresh emissions due to partial transformation of NOx to additional oxidized N products (e.g. PAN), as well as isotopic exchange between  $NO_x$  and these oxidized species; both processes will leave 15N depleted in  $NO_x$  (relative to 14N) and 15N enrichment in PAN (Walters and Michalski, 2015). If PAN existed at significant concentrations that were 1) comparable with  $NO_x$  in the atmosphere at the time, and 2) completely collected in the permanganate solution, then the  $\delta^{15}$ N-NO3- would reflect the overall  $\delta^{15}$ N of NOx + PAN in the final reduced permanganate solution. In this case, we would expect that aged smoke would not shift from the  $\delta^{15}$ N-NOx range of young smoke, because  $\delta^{15}$ N shifts in both PAN and NOx could offset each other. However, our observed  $\delta^{15}$ N-NOx mean values for both aged daytime and nighttime smoke are significantly (p<0.05) lower than that of the young smoke (shown in the figure below). This  $^{15}$ N depletion in collected "NOx" indicates the NOx in aged smoke was the predominant N species collected in the permanganate impinger during our field campaign. Similar analysis was also discussed by Miller et al. (2017). Therefore, we do not find significant isotopic evidence that PAN interferes with NOx for  $\delta^{15}$ N characterization in our study conditions.

Most importantly, we appreciate you pointing out the ground measurement of PAN in Boise in 2017 by

your group, which shows PAN of  $1.22\pm0.72$  ppbv for smoke days vs  $0.74\pm0.39$  ppbv for non-smoke days. We didn't reference it in our response to Dr. Roberts' comment because we thought there was no direct PAN measurement during our measurement period in the same environment for the same fire smokes and ages. In addition, we notice that in Figure 3 (b) (McClure and Jaffe, 2018), there are some overlaps of PAN/NOy between non-smoke and smoke periods. There are multiple times showing PAN/NOy < 0.1, with the extreme case of PAN/NOy < 0.04. As it has been pointed out in numerous publications from your group as well as others, fire emissions and the evolution of each species can have big heterogeneity depending on the fire conditions and meteorological conditions. Although PAN/NOy < 0.1is less probable, we could not rule out the possible conditions with low interference of PAN with NOx.

Indeed, we agree that correction of  $NO_x$  for PAN will yield more accurate  $NO_x$  isotopic composition. This is something we have been actively pursuing (see below). However, it is impossible to accurately quantify the interference of PAN on  $NO_x$  isotopic composition without simultaneous PAN concentration measurement, given the high temporal and spatial heterogeneity of both emissions and chemical evolution.

We believe that our work, as the first isotopic investigation of real-world wildfire derived reactive nitrogen, opens multiple channels for improvements via collaboration with other research groups including yours.

Particularly, we deem this helpful discussion as a unique opportunity and an effective way for us to improve the quantification of reactive nitrogen isotopic composition, interpretation of the isotopic results, and eventually the characterization of reactive nitrogen chemistry of not just wildfire smokes, but also other environments. A progressive approach has been designed and partially practiced:

1. In this summer's field study for quantifying the reactive nitrogen isotopic composition in an urban setting (Detroit, MI), in addition to deploying the collection system for offline isotopic analysis and a real-time  $NO_x$  analyzer (chemiluminescence), we added another  $NO_x$  analyzer connected with a HONO scrubber, and a real-time  $NO_2$  analyzer (absorption spectroscopy). This allows us to quantify real-time  $[NO_z]$ -[HONO], which includes PAN, during each of our sampling periods.

2. We are developing a 0-D box model comprised of a nearly complete reactive nitrogen mechanism, with a set of comprehensive isotopic fractionation mechanisms based on kinetics and thermodynamics. This will allow for quantitative correction of the  $NO_x$  collection technique for PAN under different product distribution scenarios.

3. Given your group's expertise in PAN measurement, we would be very interested to join in on future field measurements of (but not limited to) wildfire smokes. This will provide an essential opportunity to improve our  $NO_x$  isotopic composition quantification, particularly in environments where PAN can be important.

We would like to reiterate we hope to benefit from the helpful discussion with you and Dr. Roberts as well as the entire atmospheric chemistry community. Thank you!

Regards, Jiajue Chai

**References:**

Walters, W. W. and Michalski, G.: Theoretical calculation of nitrogen isotope equilibrium exchange fractionation factors for various NOy molecules, Geochim. Cosmochim. Acta, 164, 284–297, https://doi.org/10.1016/j.gca.2015.05.029, 2015.

Miller, D. J., Wojtal, P. K., Clark, S. C., and Hastings, M. G.: Vehicle NOx emission plume isotopic signatures: Spatial variability across the eastern United States, J. Geophys. Res. Atmospheres, 122, 4698–4717, https://doi.org/10.1002/2016JD025877, 2017.

McClure, C. D. and Jaffe, D. A.: Investigation of high ozone events due to wildfire smoke in an urban area, Atmospheric Environment, 194, 146–157, https://doi.org/10.1016/j.atmosenv.2018.09.021, 2018.

---

## Author Response (AR1)

Response to Comment and Reviews:

The response to comments and reviews are given in black and the additions to the Comment are given in red.

Response to CC1

This comment from the authors of the original paper acknowledges the interference of PAN in the NOx sampling method. They point out that the $^{15}$N isotopic signature in their collected "NOx" in aged air did not change as much as they would have expected based on theoretical considerations. This is indeed an unresolved question given that we know it was extemely likely that substantial amounts of PAN were present in the aged air masses, and other NOx photochemistry happening that needs to be considered. The resolution of this conundrum is well beyond the scope of this Comment. Instead, I have added the following concluding sentences:

The community engaged in $^{15}$N isotopic analysis appears to be left with a conundrum: why don't $^{15}$N signatures, from NOx + PAN and other reactions of NOx, match their current understanding of the effects of photochemistry? As with many such situations, this is an opportunity to learn and refine our understanding of $^{15}$N cycling in atmospheric photochemistry.

Response to RC1

I thank Dr. Jaffe for his supportive comments and for detailing his group's measurements near Boise, ID during the 2017 fire season. I have now noted those measurements in the revised Comment:

There are ground-level measurements of PAN and oxides of nitrogen in the Boise, Idaho urban area during the 2017 WF season (McClure, and Jaffe, 2018), and they show that PAN levels are a substantial fraction of odd-nitrogen, and are certainly significant relative to NOx.

There do not appear to be any other aspects of Dr. Jaffe's review that require a response on my part.

Response to RC2

I thank the reviewer for these supportive comments. As noted above, I have added concluding remarks concerning the lack of agreement of the $^{15}$N measurements with expectations given the importance of other NOx photochemistry in this environment.